# Characterization of Sodium Channel Peptides Obtained from the Venom of the Scorpion *Centruroides bonito*

**DOI:** 10.3390/toxins16030125

**Published:** 2024-03-01

**Authors:** Rita Restano-Cassulini, Timoteo Olamendi-Portugal, Lidia Riaño-Umbarila, Fernando Z. Zamudio, Gustavo Delgado-Prudencio, Baltazar Becerril, Lourival D. Possani

**Affiliations:** 1Departamento de Medicina Molecular y Bioprocesos, Instituto de Biotecnología, Universidad Nacional Autónoma de México, Cuernavaca 62210, Mexico; rita.restano@ibt.unam.mx (R.R.-C.); timoteo.olamendi@ibt.unam.mx (T.O.-P.); fernando.zamudio@ibt.unam.mx (F.Z.Z.); gustavo.delgado@ibt.unam.mx (G.D.-P.); baltazar.becerril@ibt.unam.mx (B.B.); 2Investigadora por México CONAHCYT, Instituto de Biotecnología, Universidad Nacional Autónoma de México, Cuernavaca 62250, Mexico; lidia.riano@ibt.unam.mx

**Keywords:** antivenoms, Cbo1–Cbo5, *Centruroides bonito*, scorpion toxin, single-chain antibodies, voltage-gated sodium channels

## Abstract

Five peptides were isolated from the venom of the Mexican scorpion *Centruroides bonito* by chromatographic procedures (molecular weight sieving, ion exchange columns, and HPLC) and were denoted Cbo1 to Cbo5. The first four peptides contain 66 amino acid residues and the last one contains 65 amino acids, stabilized by four disulfide bonds, with a molecular weight spanning from about 7.5 to 7.8 kDa. Four of them are toxic to mice, and their function on human Na^+^ channels expressed in HEK and CHO cells was verified. One of them (Cbo5) did not show any physiological effects. The ones toxic to mice showed that they are modifiers of the gating mechanism of the channels and belong to the beta type scorpion toxin (β-ScTx), affecting mainly the Nav1.6 channels. A phylogenetic tree analysis of their sequences confirmed the high degree of amino acid similarities with other known *bona fide* β-ScTx. The envenomation caused by this venom in mice is treated by using commercially horse antivenom available in Mexico. The potential neutralization of the toxic components was evaluated by means of surface plasmon resonance using four antibody fragments (10FG2, HV, LR, and 11F) which have been developed by our group. These antitoxins are antibody fragments of single-chain antibody type, expressed in *E. coli* and capable of recognizing Cbo1 to Cbo4 toxins to various degrees.

## 1. Introduction

Scorpions are arthropods distributed worldwide, except on the poles and some ocean islands, and are represented by 2798 different species, classified into 23 different families (https://www.ntnu.no/ub/scorpion-files) (accessed on 27 November 2023) [1].

In some regions of the world, scorpion stings can cause lethal effects on humans, as revised by Chippaux and Goyffon [2,3]. The most dangerous species belong to the family *Buthidae*, among which are the genera *Androctonus*, *Leiurus*, *Buthus*, *Buthotus*, and *Heterometrus*, which are geographically distributed in the Old World, mainly North Africa, the Middle East, and India [4]. The genus *Parabutus* is found in South Africa [5], whereas the genus *Centruroides* is located in the Southern part of the USA, Mexico, and Central America [6,7,8]. Brazil, among other countries of South America, like Argentina, Colombia, and Venezuela, have the genus *Tityus* as a dangerous species [7,8]. The venomous gland of scorpions contains many active components: enzymes, peptides, nucleotides, lipids, mucoproteins, biogenic amines, heterocyclic components, and a variety of unknown substances [9,10]. Thus far, the scorpion venom components best studied are the polypeptides that recognize ion channel receptors in excitable membranes and are harmful to different organisms, including humans [11,12,13,14,15,16]. Different families of toxins have been described, which specifically interact with ion channels of the following types: Na^+^ channels, K^+^ channels, Cl^–^ channels, and Ca^2+^ channels. It is generally accepted that the toxins of medical importance are those that affect voltage-gated Na^+^ channels since they are neurotoxins which cause nerve impulse impairment that could lead to death [17].

Mexico is home to 281 different species, from which at least 21 are potentially dangerous to humans [18,19]. In the State of Guerrero, México, 12 different species of scorpions of the genus *Centruroides* are known, among which *C. bonito* is one of the most recently described [20]. This species is known to cause human envenoming [21], but thus far, the venom of this scorpion has not been fully characterized. Here we describe the isolation of unknown peptides from this venom. The amino acid sequence of the peptides was determined, and their effect was evaluated using electrophysiological experiments on cells recombinantly expressing human Na^+^ channels. Four peptides (Cbo1 to Cbo4) were shown to affect the function of Na^+^ channels to different extents. The protecting effect of antitoxins of the single-chain antibody fragment (scFv) type generated in our group by phage-display and directed evolution is discussed. A phylogenetic analysis of these peptides is also reported.

## 2. Results

### 2.1. Venom LD_50_ Determination

The soluble venom from *C. bonito* is toxic to mice since abdominal contraction, hyperventilation, respiratory distress, cyanosis, and death have been observed. The lethal dose that kills 50% of animals (LD_50_) was determined and found to be 16.7 µg/20 g mouse body weight. For LD_50_ determination, 16 mice were injected by intraperitoneal via following the up and down method. Results are reported in the Appendix A.

### 2.2. Purification and Sequencing

In total, 80 mg of soluble venom was separated through a Sephadex G-50 column, obtaining three different well-resolved fractions (I to III, see Figure 1A). All the absorbing material from the Sephadex G-50 column was fully recovered and based on our previous experience with *Centruroides* venoms; fraction II was the one we chose to be further separated by means of an ion-exchange column containing carboxyl–methyl–cellulose resins (CMC). The CMC separation resolved 14 different fractions (fraction II–1 to II–14, Figure 1B). Fractions II–11 to II–14 were further submitted to High-Performance Liquid Chromatography (HPLC), and the corresponding profiles are shown in Figure 1C–F. The HPLC separation of fraction II–11 resolved at least 13 components (Figure 1C), from which the peptide eluting at 38.8 min was homogeneous, showed a molecular mass of 7578 ± 1 Da, and was named Cbo1. The abundance of this component estimated from the initial soluble venom corresponded to 1.7%. The HPLC of fraction II.12 allowed for the separation of nine components (Figure 1D), from which the peptides eluting at 39.30 min corresponded to 12% of the total venom. The determination of the experimental molecular mass (MW_exp_) of this component gave two ionic series corresponding to masses of 7666 ± 1 Da and 7638 ± 1 Da (Cbo2 and Cbo3). We proceeded to their sequence determination even though it was not possible to obtain the two components in separated form. The HPLC separation of fraction II–13 permitted obtaining circa 20 components (Figure 1E), from which the most abundant was eluted at around 46 min, obtained in pure form, and was denoted Cbo4. It showed a molecular mass of 7846 ± 1 Da., and it corresponded to 1.1% of the total soluble venom. Finally, HPLC separation of fraction II–14 allowed for obtaining approximately 15 components (Figure 1F), from which the component eluting at 33.1 min was homogeneous and showed a molecular mass of 7483 ± 1 Da, and this was denoted as Cbo5.

The sequences of Cbo1–Cbo5 peptides were determined by means of direct sequencing with the native and alkylated peptides, which allowed for the identification of the first 55 amino acid residues for Cbo1, the first 54 for Cbo2 and Cbo3, the first 50 for Cbo4, and the first 25 for Cbo5 (Figure 2, Edman). The C–terminal sequence was obtained after endopeptidase digestion of the carboxymethylated peptide by means of the enzymes listed in Figure 2 (AspN for Cbo1–Cbo5 plus GluC for Cbo4). The sequence of the peptide fragments resulting from digestion is indicated in Figure 2 after the enzyme name and the corresponding elution time of each obtained fragment were determined. The final sequences of Cbo1–Cbo5 were obtained by overlapping the sequence of the fragments, and they are listed in bold in Figure 2, preceded by the sequence ID and the formal name. The determination of the amino acid sequence confirmed that Cbo2 and Cbo3 sequences differ for one amino acid in position 30, where in Cbo2 there is an R30 and in Cbo3 a K30 which are both positively charged amino acids, and this is probably the reason why it was not possible to separate these two peptides.

The theoretical molecular weight (MW_th_), based on the sequence, was obtained for each peptide from the Expasy protein param tool (https://web.expasy.org/protparam/ accessed 14 March 2023) and is reported on the right side of Figure 2 (MW_th_) along with the experimental molecular weight (MW_exp_). The difference between MW_th_ and MW_exp_ of each peptide is around 8 Da that corresponds to the difference between eight cystine reduced or forming four disulfides bonds.

The toxicity of Cbo1–Cbo5 was evaluated in in vivo assays using mice. Cbo1 to Cbo4 were toxic to mice (*Mus musculus* CD1 strain), while Cbo5 peptide showed no toxic effects on mice and was further assayed in crickets (*Sphenarium purpurascens*) and sweet water shrimps (*Procambarus acanthophorus*) using doses of 20 and 50 µg per animal, where no appreciable toxic effects were observed either.

### 2.3. Phylogenetic Analysis

The amino acid sequence of the peptides from *C. bonito* was compared to similar toxins from the venom of other scorpions (Figure 3). The sequences recovered by BLASTP, and phylogenetic analysis, showed that these toxins belong to the subfamily beta of scorpion Na^+^ toxins (β–NaScTx). The amino acid sequence of Cbo1 toxin is identical to antimammalian amidated toxins Cll2b (UniProt accession P59899) from *Centruroides limpidus* [22], Cii1 (UniProt accession P59897) from *Centruroides infamatus* [23], and CpoNaTBet09 from *Centruroides possanii* [24]. The toxins closest to Cbo1 in the phylogenetic tree and with the highest percentage identity are the antimammalian toxins Co1 (UniProt accession C0HLF2) from *Centruroides ornatus* [25] and Cb1 (UniProt accession C0HLR3) from *Centruroides baergi* [26], with which Cbo1 shares 98% and 97% amino acid identity, respectively (Figure 3A,B). Toxins Cbo2 and Cbo3 share 98% amino acid identity, with Cbo3 differing by a single amino acid at position 30 relative to Cbo2 (K30R) (Figure 3B). The toxin phylogenetically closest to Cbo2 and Cbo3 is toxin Cll4 (UniProt accession Q7Z1K8) from *C. limpidus*, the antimammalian toxin Co2 (UniProt accession C0HLF3) from *C. ornatus* [25], and Chui5 (UniProt accession C0HM18) from *Centruroides huichol* [27], with which Cbo2 shares 95, 92, and 95% amino acid identity, respectively (Figure 3A,B). The Cbo4 toxin shares 98% identity with the antimammalian Cl13 toxin of *C. limpidus* [28] and 91% identity with Cb2 of *Centruroides baergi* [26] (Figure 3B). The Cbo5 toxin is in a major clade distinct from the other Cbo toxins. In the Cbo5 clade is the CsEl toxin (UniProt accession P01491) from *C. sculpturatus*, with 83% amino acid identity; this toxin preferentially affects nonmammalian vertebrates (affects channels from chicken and frog) [11,29]. Another toxin with the same percentage of similarity is Co52 (UniProt accession C0HLF8) from *C. ornatus*, a toxin with no activity against mice, chickens, crickets, or woodlice [25] (Figure 3B).

### 2.4. Physiological Characterization

The five peptides were evaluated for their activity on human voltage-gated sodium channels stably expressed in HEK (hNav 1.1 to 1.6) and CHO (hNav 1.7) cells. Currents were elicited in control condition (black traces) and in the presence of a toxin applied at a final concentration of 200 nM for 30–60 s (olive and green traces). Of the five peptides, Cbo5 was the only one that showed no activity in the sodium channel subtypes evaluated (results are shown in Appendix A). The other four peptides act as beta scorpion toxins; thus, they shift the activation threshold to more negative potential and reduce the total current in a different magnitude depending on the toxin and the channel (Figure 4). Cbo1 was shown to be selective for hNav 1.6 channels. As previously explained, toxins Cbo2 and Cbo3 have just one conservative amino acid change in position 30 (arginine in Cbo2; lysine in Cbo3). For this reason, it was almost impossible to obtain pure peptides and they were applied as a mix of both toxins at a final concentration of 200 nM. Cbo2–Cbo3 mix was weakly active on hNav1.6, where they shifted the activation threshold but did not significantly reduce the peak current. In the hNav 1.4 channels, the current reduction was also not significant, as well as the activation shift. All the other channels were insensitive to the Cbo2–Cbo3 toxin mix. On the contrary, Cbo4 is promiscuous, affecting several sodium channel subtypes. It significantly modifies the gaiting properties of hNav 1.4 and hNav 1.6, shifting the activation process and reducing the peak currents. Like other beta scorpion toxins [30], Cbo4 reduces the peak current of the hNav 1.5 without affecting the activation threshold. Moreover, Cbo4 slows down the inactivation process in hNav 1.5 and hNav 1.6. Cbo4 also slightly acts on hNav 1.1 and hNav 1.2, where modifications on the activation threshold and peak current were present, but at a 200 nM toxin concentration, these modifications were not statistically significant (at *p* = 0.05 level), and no apparent activity was observed in hNav 1.3 and hNav 1.7.

### 2.5. Neutralization with Single-Chains

In our group, following the interest of generating a protective antidote, we have obtained a series of neutralizing antibody fragments against toxins and venoms of Mexican scorpions [27,31,32,33]. In preliminary assays of protection against 1 LD_50_ of *C. bonito* venom (without knowing the characteristics of the toxins and their proportion in this venom), we observed a delay in the appearance of envenoming signs, with respect to the control group (1 LD_50_ of venom), where the initial signs of envenoming are sensitivity to noise and uncontrolled movements. At 30 min after the venom injection, oscillatory movement of the tail and cyanosis was observed. Subsequently, respiratory distress was evident. Finally, death occurred in a timelapse of 2 or 4 h. The animals treated with 1 LD_50_ of venom and the mixture of single-chain variable fragments (scFvs) LR and 10FG2 were able to survive.

The appearance of the described signs was not observed. However, there was a delay in the onset of severe respiratory distress, suggesting that there was a partial neutralization of the venom. To explain these results, we performed real-time analyses of the interactions of scFvs with the identified toxins of *C. bonito* (see sensograms, Figure 5). We found that Cbo toxins 1–3 were not recognized by scFv LR, whereas they were well recognized by scFv 10FG2. Considering these results, it is likely that 10FG2 neutralizes them. The Cbo4 toxin was not well recognized by either LR or 10FG2, which explains the appearance of the envenoming signs. Recently, new antibody fragments have been developed, such as scFv HV [27], which shows good interaction with Cb2 and Cbo3 toxins. Its low dissociation from the toxins is remarkable. In the case of the Cbo4 toxin, there is no good interaction with the scFvs evaluated, and the only one that does interact is scFv 11F, although it showed fast dissociation.

## 3. Discussion

Since *C. bonito* is a scorpion species which has recently been described and its venom contains toxic components to humans, we decided to obtain venom from these animals, aiming to conduct formal studies of its components. The venom components separated by chromatographic procedures were assayed in vivo using mice. Several components (Cbo1, Cbo4, and Cbo5) were obtained in pure form, while for two components (Cbo2 and Cbo3), it was impossible to separate them due to the high similarity in their sequences; in fact, they differ by one conservative change (R30K). The full amino acid sequences were obtained by using Edman degradation and enzymatic digestions. The molecular weights found by mass spectroscopy correspond to those expected from the full sequences obtained with a difference of about 8 Daltons (8 D), which explains the oxidized state of the cysteines. Phylogenetic analysis performed with this information and with known scorpion toxins from venoms of other *Centruroides* species suggested that these *C. bonito* venom components belong to another known beta type of sodium scorpion toxins (β–NaScTxs). This analysis showed two strongly supported clades (branch support of 90) in which *C. bonito* and related β–NaScTxs are clustered (Figure 3). The clade in which the Cbo5 toxin is grouped comprises other toxins with diverse activities, including those affecting insect–crustaceans, such as Cn1 from *Centruroides noxius* [34,35], vertebrates, such as CsE1 [11,29], and toxins with functions not yet elucidated (e.g., Co52, Ct17, and others) [25,36]. Like the latter, Cbo5 has no detectable effect on sodium channels from human origin, as well as in in vivo experiments on mice, crickets, and sweet water shrimps. These negative results suggest that these components may act on a different type of receptor or on a different type of organism [22,37,38]. In a similar way, the poor branch support observed in Cbo5 (<50) could be explained by a limited sampling of the sequences available in the databases; this effect can be observed in the percentage identity shared by Cbo5 with other nearby toxins, which do not exceed 83% amino acid identity (Figure 3B). The second clade includes phylogenetically more recent toxins, including toxins Cbo1 to Cbo4 and other antimammalian toxins. Toxins such as Cbo1, whose sequence has been reported in other species of the genus *Centruroides* (see Figure 3), may have been present in common ancestors and preserved during the speciation process of the *Centruroides* genus, which appeared at the end of the tertiary period of the cenozoic period when mammals, potential predators of these animals, diversified [18]. Cbo1 and Cbo2–Cbo3 have an effect on Nav1.6 channels. This confirms the in vivo assays performed with mice. Cbo4 is the most active toxin, modifying the function of several channels, mainly hNav.1.4, hNav1.5, and hNav1.6. This toxin is highly similar to Cl13 (they differ in position 5, where Cbo4 has the amino acid I and Cl13 has L) and, as expected, they have shown very similar effects in voltage-gated sodium channels [39]. Moreover, it is important to mention that due to the sensibility of the equipment utilized for mass determination, it was not possible to show if the Cbo toxins here described have been amidated in the C–terminal like many toxins of the same group (see multiple alignment in Figure 3B).

From the characterization of the toxic components of any given venom, it is possible, through scFv affinity maturation, to implement rational strategies to achieve a good neutralization of each of the toxic components [40,41,42,43]. The scFvs 10FG2 and HV evaluated in this work show a good level of interaction with the evaluated toxins similar to those that have already been neutralized [27,33]. These scFvs could contribute to the neutralization of toxins Cbo1, Cbo2, and Cbo3, which are important toxic components of the venom (together they correspond to 13.7%). In the case of Cbo4 toxin, which is also an important component (1.1% of the venom), we do not yet have an scFv that shows a good interaction which could contribute to the neutralization of the venom. The scFv 11F is a good candidate to be matured in vitro, and then neutralization of this toxin could be achieved. These results are promising since three of the four toxins are well recognized, and there is a good chance that neutralization of the entire venom will be achieved relatively soon.

## 4. Materials and Methods

### 4.1. Venom Extraction, LD_50_ Determination, and Purification

Venom was obtained from approximately 500 animals collected in the State of Guerrero (region of Costa Chica) by means of electric stimulation (SEMARNAT, official permission SCPA/DGVS/00367/22). The venom was dissolved in water, centrifuged at 15,000× *g* for 15 min, and the supernatant was lyophilized and kept at −20 °C until use. The lethal dose 50% (LD_50_) of whole venom was estimated using 16 mice (*Mus musculus* strain CD1) by intraperitoneal via following the method of up and down described in [44]. The use of animals in this work complies with the guidelines established by Mexican legislation and internationally recommended procedures and was approved by the Bioethics Committee of the Instituto de Biotecnología of Universidad Nacional Autónoma de México, number 413.

Initial purification was performed using 80 mg of *C. bonito* soluble venom separated in a Sephadex G-50 column (2 m high, 0.9 cm diameter) run in the presence of 20 mM ammonium acetate buffer, pH 4.7. From our previous experiences with *Centruroides* venoms of different species, we have learned that fraction II of Sephadex is the one that contains most of the venom toxic component. With this idea, Fraction II was further separated by ion-exchange chromatography on a carboxy–methyl–cellulose (CMC) column (5 mL volume) run in the same buffer and eluted with a sodium chloride gradient from 0 to 0.5 M salt, during 1000 min at 0.5 mL/min, in a New Generation Chromatography (NGC^TM^) system connected to a fraction collector (both from BioRAD, Mexico City, Mexico). Fractions from the CMC column were further separated through high-performance liquid chromatography (HPLC) using a Waters model 1525 binary HPLC pump, detector 2489 UV/vis chromatographer (Mexico City, Mexico) in the conditions earlier described [45]. Basically, an analytical C18 reverse-phase column from Vydac (Hisperia, CA, USA) was used (C18 250 × 4.6 mm, wakopac/wakosil pTH–II 4.6 × 250 mm). The components were eluted with an acetonitrile gradient from 0 to 60%, and the mobile phase was prepared with solution A, containing 0.12% trifluoroacetic acid (TFA) in water, and solution B contained 0.10% TFA in acetonitrile. For each component, the columns were run for 60 min at a 1mL/min flow rate. Our main interest is to find the venom components principally responsible for the venom intoxication and, in *Centruroides* venom, these are mainly the sodium channel toxins. Our first approach was to select the most abundant peaks obtained from HPLC and those eluting after 30 min because we empirically know that in this interval, the sodium channel toxins should be eluted. The relative concentration of the fractions was estimated from the chromatogram, dividing the area of each peak by the sum of the area of all the peaks. The chosen components were analyzed for purity by mass spectrometry analysis and sequencing determination by Edman degradation, as mentioned below. Toxicity to mammals of the sequenced peptides (Cbo1–Cbo5) was evaluated using two mice (*Mus musculus* CD1 strain) injected with 10 µg/20 g mouse weight. Toxin Cbo5 was also assayed in crickets (*Sphenarium purpurascens*) and in sweet water shrimps (*Procambarus acanthophorus*) using the technique earlier described [26]. For these experiments, 2 crickets were injected with 20 µg and 2 were injected with 50 µg of Cbo5, while 2 sweet water shrimps were assayed with 20 µg of toxin.

### 4.2. Mass Spectrometry and Sequence Determination

The purified fractions were reconstituted to a final concentration of 500 pmol/5 μL of 50% acetonitrile with 0.1% acetic acid and directly applied into a LCQFleet ion trapp mass spectrometer (Thermo Fisher Scientific Inc. San Jose, CA, USA) using a Surveyor MS syringe pump delivery system. The eluate at 10 μL/min was split to allow for only 5% of the sample to enter the nano spray source (0.5 μL/min). The spray voltage was set from 1.0 to 2.0 kV, and the capillary temperature was set from 100 to 200 °C. All spectra were obtained in positive-ion mode. The equipment used for this analysis has a precision of ±1 Da when using peptides of about 7 to 8 kDa. The purified peptides were subjected to Edman degradation in a Shimadzu Protein Sequencer PPSQ–31A/33A (Columbia, MD, USA). Homogeneous peptides were analyzed in two formats: native peptide and reduced and alkylated cysteines with iodoacetic acid for the purpose of determining their position in the sequence. In our conditions, most pure peptides in their native format allow for the identification of approximately the first 50 amino acid residues, starting from the N–terminal amino acid. In order to obtain the full sequence, the reduced and alkylated format of the peptides was submitted to enzymatic hydrolysis [45]. The corresponding peptides after treatment with the endopeptidases (GluC and AspN) were subjected to HPLC separation, and their sub-peptides were placed into the sequencer. Both enzymes, from the company Roche Diagnostics GmbH (Mannheim, Germany), were used according to the protocol suggested by the provider. The overlapping of the sequences and the verification of the completeness of the sequence were confirmed by mass spectrometry determination of the pure native peptides and compared to the expected molecular weight based on the amino acid structure experimentally determined. The amino acid sequence of the five peptides was registered at UNIPROT and was given the following identification numbers: Cbo1 is C0HMA3; Cbo2 is C0HMA4; Cbo3 is C0HMA5; Cbo4 is C0HMA6; and Cbo5 is C0HMA7.

### 4.3. Phylogenetic Analysis

The search for potential homologs of *C. bonito* toxins was performed using BLASTP v2.13.0+ [46] in a command line. All 569793 proteins listed in UniProtKB/Swiss–Prot Release 2024_01 (UniProt, 2023) (accessed on 26 September 2023) were used as a database. Each of the *C. bonito* sequences was used as a query using an *evalue* = 1 × 10^–20^ as the significance limit. Redundant UniProt IDs from BLASTP analysis were removed using unix commands. An additional BLASTP search was performed in the NCBI nr database. The sequences of the mature chains of 56 scorpion Na^+^ toxins from the beta subfamily (β–NaScTx) recovered from BLASTP and 3 Na^+^ toxins from the alpha subfamily (α–NaScTx) (outgroup) were aligned with MAFFT v7.490 [47] using the progressive FFT–NS–2 method. Construction of an initial phylogenetic tree was performed by Bayesian inference using MrBayes v3.2.7 [48] with the options *lset rates* = *gamma* with *prset aamodelpr* = *fixed*. The WAG model [49] was chosen as the amino acid substitution model. The analysis was run for 1 × 10^7^ generations in eight chains with a sampling frequency of every 1000 trees. During the construction of the consensus tree, the first 1 × 10^6^ trees were discarded (Burn-in). The initial tree was pruned by eliminating clades poorly related to the *C. bonito* sequences. A new phylogenetic analysis by Bayesian inference was performed using 41 remaining β–NaScTx; the 3 α–NaScTx retained as an outgroup. Percentage identity and amino acid similarity between sequences were determined with “The sequence manipulation suite” [50].

### 4.4. Cell Culture and Electrophysiology Characterization

Cell culture and the electrophysiology experiment were performed according to the protocols described in [26]. Briefly, HEK cells expressing human voltage-gated sodium channels of the subtypes 1.1 to 1.6 (hNav 1.1–hNav 1.6) and CHO cells expressing hNav 1.7 were cultivated in Dulbeco’s Modified Eagle Medium supplemented with 10% fetal bovine serum and 500 µg/mL of antibiotic G418 and were maintained at 37 °C and 5% CO_2_ in a humidified atmosphere. Cells were kindly gifted from Prof. Enzo Wanke lab (University of Milan-Bicocca, Italy). Currents were elicited by depolarizing steps from −120 to 40 mV with 10 mV increment, preceded by a strong and short depolarization (5 ms at +50 mV), and then they were tested at their full activation potential (−30 or 10 mV). For the activation and inactivation curves, current and conductance were plotted against the corresponding membrane potential. Data were fitted with single or double Boltzmann equation [51] and are presented as a mean of 3–7 experiment ± standard deviations.

### 4.5. Evaluation of the Interaction of scFvs and C. bonito Toxins by Surface Plasmon Resonance

The scFv variants were tested against *C. bonito* toxins in a biosensor that detects molecular interactions in real time (Biacore X100, Uppsala, Sweden). Each toxin was dissolved in 10 mM 2–(N–morpholino) ethanesulfonic acid (pH 6) and immobilized on cell 2 of a CM5 sensor chip using the amino coupling kit, reaching binding levels of 200 RUs. Cell 1 in the sensor chip without antigen was used as a control. The scFvs LR, 10FG2, HV, and 11F [27,31,32,33] were used to evaluate binding to each toxin. The samples were diluted in HBS–EP buffer (Biacore) at 100 nM concentration, and 100 μL of the scFvs was injected over the chip at a flow rate of 50 μL min^−1^ at 25 °C with a delay time of 500 s. The chip surfaces were regenerated with 10 mM glycine–HCl pH 2. The sensorgrams were corrected by subtracting the values from the reference flow cell and comparing the results using BIA-evaluation software version 3.1.

## Figures and Tables

**Figure 1 toxins-16-00125-f001:**
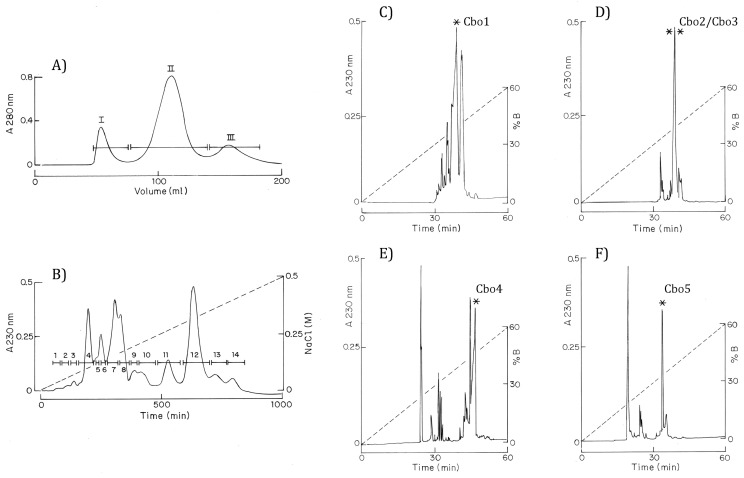
Purification of *C. bonito* peptides. (**A**) Separation of 80 mg of soluble venom by means of Sephadex G-50 column. (**B**) CMC separation of fraction II from Sephadex G-50 column. A column of 5 mL capacity was equilibrated and run with 20 mM ammonium acetate buffer pH 4.7. The elution was performed with a linear gradient of 0 to 0.5 M sodium chloride, during 1000 min at 0.5 mL/min. Fractions II–11 to II–14 were further separated by HPLC. The peptides of interest are indicated with an asterisk (*) and correspond to the following factions: (**C**) II–11–38.8 (Cbo1), (**D**) II–12–39.30, and 39.31 (Cbo2 and Cbo3) that correspond to the ascendant and descendant section of the indicated peak, both containing Cbo2 and Cbo3. Panels (**E**,**F**) correspond to fractions II–13–46.8 (Cbo4) and II–14–33.1 (Cbo5), respectively.

**Figure 2 toxins-16-00125-f002:**
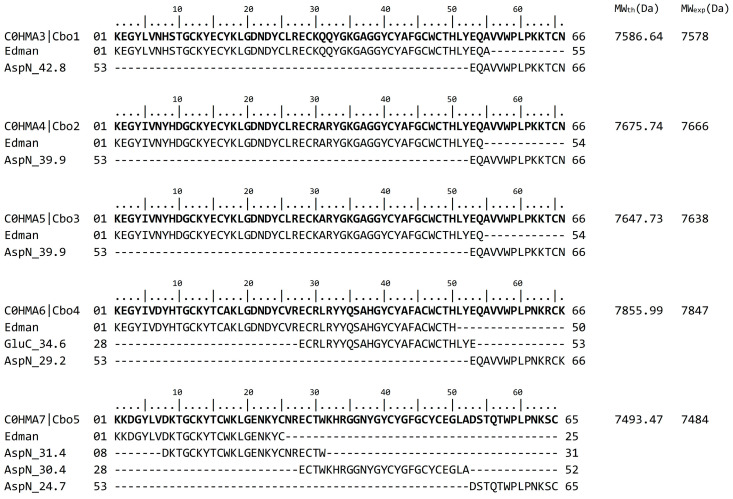
Amino acid sequence obtained from the Edman degradation of the fractions highlighted with an asterisk (*) in Figure 1. The first line of each one of the 5 peptides/toxins contains the full sequence obtained (listed in bold). The second line shows the sequence of the same sample reduced and alkylated (indicated as Edman). The subsequent lines in each toxin are the sequences obtained from the peptides after enzymatic digestion, necessary to obtain the entire sequence by means of overlapping the corresponding fragments. The time of elution from the HPLC of each digested peptide is indicated on the left after the name of the enzyme. Numbers at the left and the right of the sequences indicate the first and the last position according to the length of the complete sequence. On the right side of the figure, the theoretical molecular weight (MW_th_) calculated from the given sequences and the experimental molecular weight (MW_exp_) obtained in the ESI–MS determination are reported.

**Figure 3 toxins-16-00125-f003:**
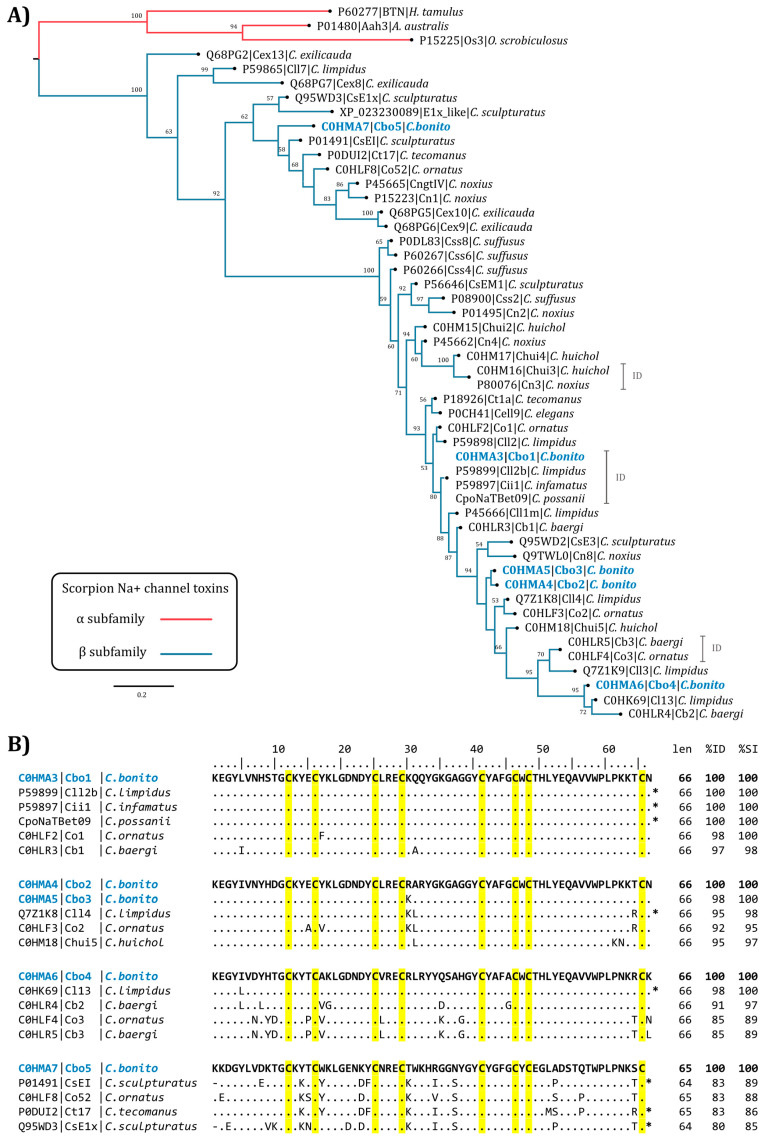
Phylogenetic analysis and multiple alignment of *C. bonito* toxins and other related β–NaScTx. (**A**) Tree topology obtained from Bayesian analysis of Cbo1–5 toxins and other 35 related β–NaScTx from scorpions of *Centruroides* genus. ID indicates identical sequences. The numbers below the nodes indicate percentage values of posterior probability greater than 50. The scale bar represents the number of amino acid substitutions per site. Analysis was performed using mature chains. Sequence names are composed of the accession code UniProt, followed by the toxin name and the species name. Sequences recovered from the NCBI are indicated with their accession numbers, followed by toxin name and the species name. CpoNaTBet09 was retrieved from the original publication [24]. Three α–NaScTx (BTN, Aah3, and Os3) were used as outgroups and to root the tree. (**B**) Multiple alignment of *C. bonito* toxins and other β–NaScTxs from scorpions of *Centruroides* genus. UniProt sequences closely related to Cbo1, Cbo2, Cbo3, Cbo4, and Cbo5 are shown. Len indicates mature chain length; %ID indicates percent amino acid identity. %SI indicates the percentage of amino acid similarity. Conserved cysteine residues are highlighted in yellow. Identical positions to *C. bonito* toxins are indicated by dots. (*) indicates toxins where the C–terminal amidation has been reported. The UniProt access code is shown on the left, followed by the toxin name and species name.

**Figure 4 toxins-16-00125-f004:**
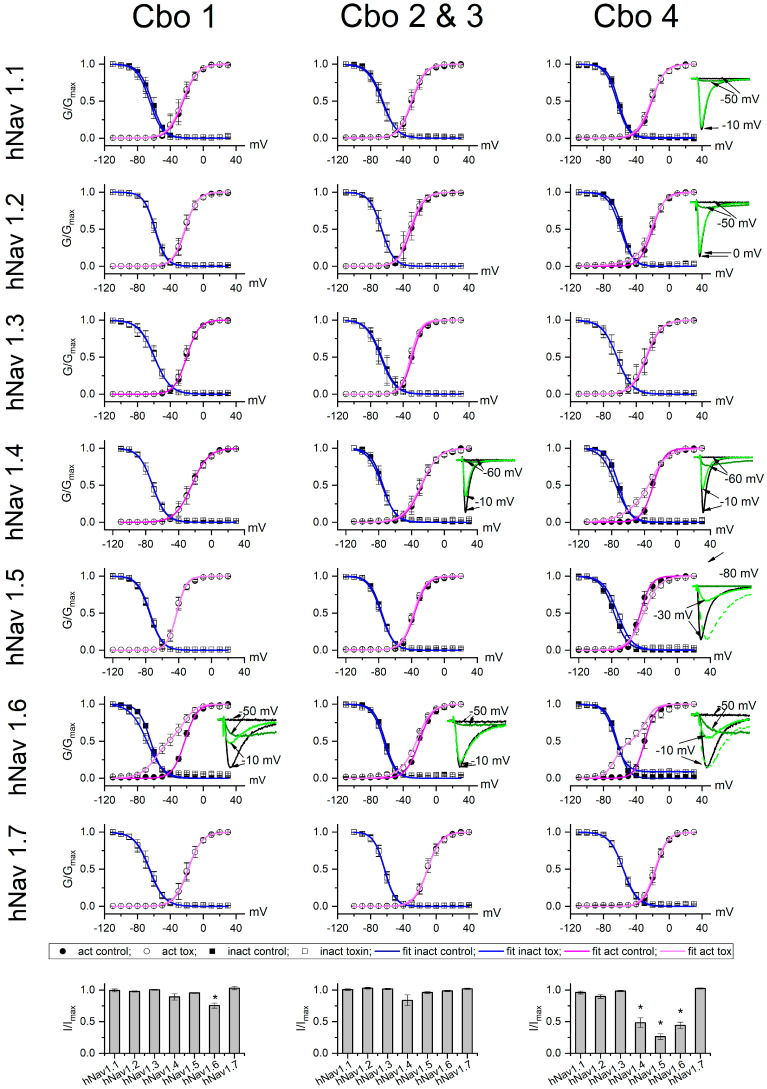
Activity of Cbo1, Cbo2 plus Cbo3, and Cbo4 toxins on human voltage-gated sodium channels hNav 1.1–hNav 1.7. The figure shows the curves of activation (circles) and inactivation (squares) processes of hNav 1.1 to hNav1.7. Data were obtained in control conditions and during application of toxin at 200 nM. Lines are the best fit obtained by a Boltzmann equation or the sum of two Boltzmann equations when toxin modifies the voltage dependence of activation process. This was the case of hNav 1.6 with Cbo1–Cbo4 and hNav 1.2 and 1.4 with Cbo4. Cbo4 is also capable to slowing down the inactivation process in channels hNav 1.5 and hNav 1.6 as shown here. Representative traces of currents affected by the toxins are shown in the small panels, where black traces are currents in control recorded at sub-threshold and full-activation potentials; in olive and green are sub-threshold and full-activated currents with toxin. Dashed traces in green are currents recorded at –30 mV for hNav1.4 or –10 mV for hNav 1.5 and scaled to the control traces to better appreciate the inactivation delay. Symbols and line colors correspond to the legend. The effect of toxins on the peak currents is graphed in the bottom row. Data are the mean of 3–7 experiments ± standard error. The asterisk (*) indicates significance at 0.05 level of the difference between control and toxin conditions by means of a “Paired Sample t Test”. The activation and inactivation fitting parameters, along with the fractional residual current values, are reported in Appendix A.

**Figure 5 toxins-16-00125-f005:**
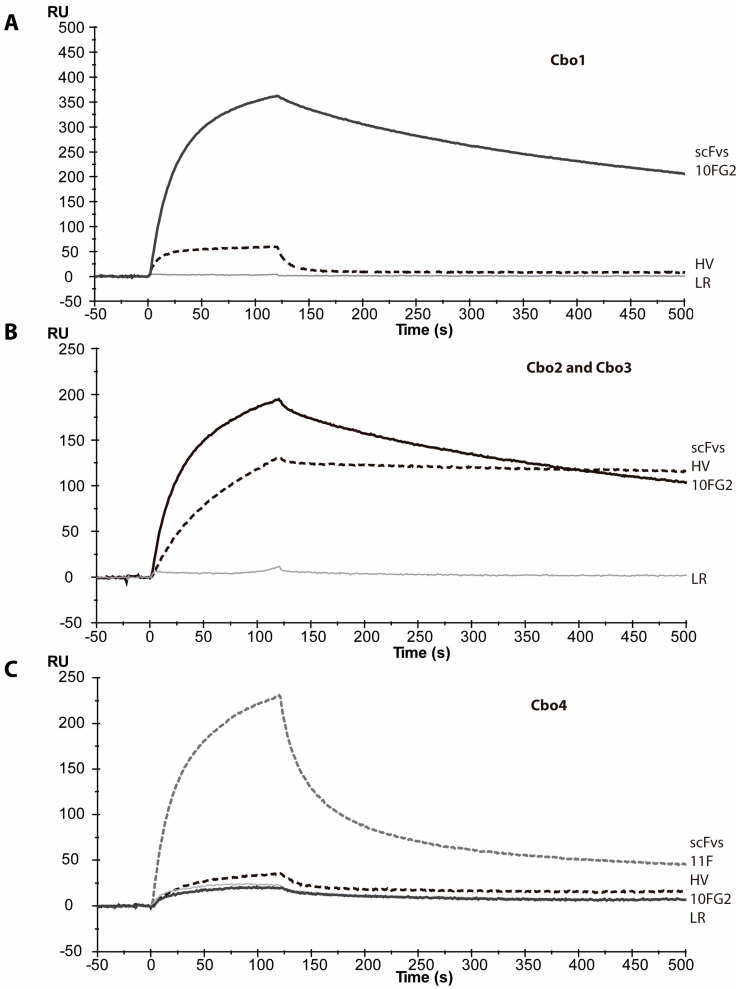
Sensorgrams of interaction between scFvs (at 100 nM) and toxins (**A**) with Cbo1 toxin, (**B**) with Cbo2–Cbo3 mix, and (**C**) with Cbo4 toxin. Biosensor measurements were performed at 25 °C with a flow rate of 50 µL/min. RU stands for resonance units, a standard parameter for surface plasmon resonance analysis. 11F, HV, 10FG2, and LR are the arbitrary names for the single-chain variable fragment (scFvs).

## Data Availability

The data presented in this study are available on request from the corresponding author.

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
