# Peer review of "Characterization of Sodium Channel Peptides Obtained from the Venom of the Scorpion Centruroides bonito"

_toxins, 2024, doi:10.3390/toxins16030125_

Round 1

Reviewer 1 Report

Comments and Suggestions for Authors

The manuscript titled “Isolation and characterization of five new peptides that affect Na+ -channels function, from the venom of the scorpion Centruroides bonito describe the new peptides were isolated from scorpion toxins and selectively activate sodium channels. toxicity of the peptide toxin in mice was inhibited by preparing neutralizing antibodies. The content of this manuscript represents a valuable original research work suitable for publication in Toxins. Minor errors should be corrected, and some questions need to be addressed before finalizing the publication of this manuscript.

1. The description of the previous findings and the findings of this article is a bit confusing, so that readers can not well distinguish which are the findings of this article.

2.  Fig.4 results show that Cbo1 selectively inhibits human Nav1.6 and Cbo2&3 are insensitive to Na+ channels. However, these peptides are toxic to mice, has there been any research on the sensitivity of these peptides to Na+ channels in mice?

3.  Do these peptide toxins isolated in this paper cause toxicity in mice through Na+ channels?

4.  Can scFv HV neutralize Cbo2&3-induced poisoning in mice? Can the combination of scFvs 10FG2 and scFv HV completely neutralize Cbo2&3-induced poisoning in mice?

Comments on the Quality of English Language

Line-57 “80 milligrams”  changed to “mg”

Reviewer 2 Report

Comments and Suggestions for Authors

A brief summary 

The Paper “Isolation and characterization of five new peptides that affect Na + -channels function, from the venom of the scorpion Centruroides bonito” wants to describe 5 novel venom toxins. Therefore the authors tried to isolate the peptides out of the crude scorpion venom and characterized them afterwards in regards to amino acid sequence and physiological parameters. The general procedure follows an understandable workflow, but while reading several critical points appear in the argumentation and experiment structure, that has to be clarified by the authors.

There I suggest to reject this manuscript for now. If the authors can extend the experiments and deliver missing information (in some cases it is just not clear, what has been done for which samples, like the animal tests on crickets, or are missing like proper MS data presentation), I would suggest a new resubmission in future.

General concept comments

In the following I want to highlight some of the most critical points to me, while a detailed list follows later:

Even so the authors state in the title that five new peptides has been isolated, the text shows that one has already three known identical sequences (Cbo1, Figure 3B) and two could not be separated (Cbo2&3, line 259), which makes it also hard to understand how then the sequences could be obtained. And with this, the in line 16-18 stated antibody tests against “purified toxins” is not correct.

The author don’t provided any validation of mass spectrometry (MS) data, no original data are provided nor shown anywhere else, as later commented in detail. Further highly critically point:

line 287-290: sensibility of the equipment is mentioned to be not efficient for the detection of amidation (~1 Da), but previously the observed to theoretically mass are given on 1 digit behind the comma. So either the resolution of the MS is high enough to calculate the mass that precise, but then an amidation should be visible, or if not that precise: the whole part about the mass determination needs to be rewritten.

The authors mention in line 9 (abstract) the four disulfide bridges, but except the sequences showing eight Cys, no validation of the bridges were made, like MS confirmation of reduced versus non reduced peptide masses.

The given Uniprot IDs of all five peptides are not accessible at the Uniprot webpage (15.12.2023), and could be controlled. Additionally, the peptide sequences should be at least in SI set at fasta file, since in the main body they only are accessible in an image - and manually copying 5 times 50 amino out of an image is unnecessary complicated, which makes a revision complicated.

The animal tests operate with n=2, which is way too low for any confident statistic or LD50 determination, or it’s not clearly mentioned in the Materials, since the authors refer to other studies instead of mentioning the parameters in this manuscript. Furthermore, line 256 mentioned “mice or crickets and sweet water shrimps” has been used, but no figure or table show those results, nor it is defined which cricket or sweet water shrimps species was used. It is not clear, which peptides were tested against which animals, since the Materials (line 312) only mention crickets and shrimps for Cbo5

Specific comments

Direct comments to the figures:

Figure 1.

A) the runtime of 1000 min fits not to the axis, which shows ml, I highly recommend to make it equal to the other images an set time on the x axis.

B-E) Legend don’t mention all five peptides, nor are five asterisked peaks of interest marked.

Figure 2. Description of a line should not be placed in the amino sequenced, but on the left outside of the sequence body (e.g. Asp N 42.8). At the moment the line identities are either missing, placed on the left or right. Furthermore, I highly recommend not to abbreviate GluC with V8, which is not a commonly used abbr. for an abbr.

Example for a better readable structure, equal to Figure 3B:

Cbo4      01 KEGY…

Edman     01 KEGY…

GluC_34.6 28 ----…

AspN_29.2 53 ----…

Figure 3. The resolution is low and should be  adjusted, so that the branch values can be read. Highlighting only one identical sequence in green adds an unnecessary highlight. The 0.2 in (A) has no value. The yellow asterisk in (B) to mark C-terminal modification might better be visible in black. In (B) not all taxa are italic

Figure 4. For a better readability the authors should avoid twice grey for the ascending as well as descending lines. It is not easy visible, how the data behave on the intersection points.

Figure 5. Please explain y axis unit “RU” and lable the subfigure A,B, C as in the other figures, for consistency.

Detailed commnents:

line 38: Please change “man” to “human” or another non-sex related word, since the venom is also toxic to woman.

line 43: The genus Centruroides (C.) has been already earlier mentioned and should be abbr. there.

line 46: “full amino acid sequence of the peptides was determined” is not correct, since the C-terminal modification could not be determines

line 48/49: Toxins are mentioned as “Cbo 1” and “Cbo4”, please check the manuscript for consistency about the space between Cbo and the index.

line 52: The introduction lacks abit more information about neurotoxins and their mode of action, as well as occurrence in animal venoms in general or for scorpions in detail.

line 55: “was toxic to mice”, please mention how this determined, like symptoms etc. Additionally, it is a bit confusing to start the “2.1. Purification and Sequencing” part with such an information.

line 58: “(I to III, not shown; see Figure 1A in reference [22])” - this information should be shown at least in the supplementary and not referred to another publication

line 60: “was  the  only  toxic  to  mice” again, how was this determined. The authors should consider a rephrasing of this paragraph.

line 63/64: “A peptide of fraction II-14 (non toxic) was also obtained in pure form.” how was this determined to be a peptide, and what is the sequence?

line 67: beside the already mentioned problems with the MS analysis, the molecular masses should be shown in a table with theoretically, observed masses including the information of reduced or non-reduced sample, either in MH+ (m/z) or molecular mass (Da). But with the technical limits on resolution mentioned and not given more decimals than reliable. Given on the claims of the authors, that an amidation can not be observed the limit would not be the ones position, but only tens digit.Therefore, as previously mentioned, the whole part about mass comparison needs to be reevaluated.

line 83/84: “The final content of this component estimated from the initial soluble venom showed that it corresponds to 1.7% of the venom” - How was it determined? Based on the 230nm signal? And if, it lacks the proof, that the isolated fraction were pure without further peptides masses. This could be done by MS1 spectra.

line 98: “7667.7” unit (Da?) missing. Is this the observed or calculated mass?

line 105: “apparatus” is an uncommon word in this context.

line 108/109: “obtaining circa 20 components” -  Can the author please describe how they identify the relevant peptide fraction for this and the other peptides. It is not clear why always one peak was selected. Was each fraction/subfraction/peak tested in all three animal to be toxic? Was the peptide content determined by MS or a protein gel, to exclude only protein containing peaks? As a reader it is not understandable what, in this example, the other 19 peaks are, and why they are not of importance.

line 114: “Glu-C, also known as V8”, I suggest to not abbr. GluC/Glu-C as V8. This is a not common abbr. for the proteinase and makes it harder to understand what has been made.

line 172 (Figure 2): The selection of sequences to compare seems a bit random, since sometime a few sequences from a lower or higher branch were taken, and then not always all from there.

line 174: “Phylogenetic analysis and multiple alignment of C. bonito toxins” header should be renamed, since not only C. bonito toxins are compared.

line 184: Maybe the authors can consider to insert beside the %ID also percentage of similarities as another point, to underline the high similarity even after amino acid change. But this is just a personal suggestion.

line 192-194: “Of the five peptides, Cbo5 was the only one that shows no activity  in  the  sodium  channel  subtypes  investigated  (results  are  shown  in  Supplementary  Figure 1)” - I would suggest to insert the data into figure 4, to have direct comparison between the active and non-active peptides. But this is just a personal preference.

line 218: “activation (circles) and inactivation (squares)” is not readable in the figure, as well as “(black symbols) and during application of toxin at 200 nM (empty symbols)” are not good to see, due to overlaying effect. The authors might consider to a figure legend, like in Figure S1

line 226: “Data  are  the  mean  of  3-7  experiment” - please mention somewhere what n is for the different tests, and why not always 7 has been perfomed.

TABLE S1: It is not clear, what the empty cells for A and I/I max are, since sometimes “n.c.” is marked. Where the others not calculated, but then why are they not “n.c.” also? Why does Cbo4 has no I/I max values.

line 231/232: “we have obtained a series of neutralizing antibody fragments against toxins” - What is the origin of those antibody fragments? Please consider to state it in the text.

line 234: please clarify the abbr. “scFvs 10FG2 and LR”

line 236: “however, clear signs of envenoming were observed”, please state which symptoms.

line 239: “Cbo toxins 1, 2 and 3”, I recommend to stay to the previous annotation of Cbo1-3 as previously.

line 241: “10FG2” since the lack of abbr. it is not clear what the difference between 10FG2 and scFvs 10FG2 is, if there is any.

line 244: “Its low dissociation” - How was this dissociation determined? (Refer to the line in the corresponding Figure 5 subfigure)

line 262: “mass spectroscopy  correspond to  those expected” this is, as stated previously, wrong since lack of precision, and correspond and expected values where never directly stated.

line 263/264: “known  scorpion toxins from venom of other Centruroides species” The author should consider to compare the sequeucn

line 285: “Nv1.6.” is this Nav1.6?

line 292: What is considered with ”new technologies”? The authors might consider the review “Modern venomics - Current insights, novel methods, and future perspectives in biological and applied animal venom research “ (https://doi.org/10.1093/gigascience/giac048

line 297: “the most abundant toxins in the venom (together they correspond to 13.7%).” - This cannot be stated without investigate the other >85% of the venom. This includes then also the statement of line 301-303 “These results are promising since three of the four toxins are well recognized and there is a good chance that neutralization of the entire venom will be achieved relatively soon.” It is not reliable to say, based on <15% inactivated (with a quantification not more precise defined - the concentration of smaller compounds in the venom could be way higher, or other extremely potent) part, that the soon the other >85% can be soon neutralized. Especially, since human data only rely at the moment on the receptors and not whole organisms.

line 306: What is the number of milked specimen? Where they all from the same region/population?

line 308: unit “g” should be changed to “x g”

line 309: recheck the “°”, it seems underlined in the °C

line 321: Please write out the abbr. “NGC”

line 322: How where the “toxic fractions” determined?

Determination by Mass Spectrometry:

The peptides are mentioned to be analysis by MS, but neither are raw data (or processed data) given, or shown in any figure to confirm the peptide masses by MS Spectra. Those MS1 spectra should be placed at least in the supplementary part. Theoretically masses of the peptides should be calculated and compared to the MS1 masses measured, and shown at least in the supplementary as a table.

5.1 Mass spectrometry and sequence determination

line 335: ‘apparatus’ is not a commonly used phrase, and might be changed.

line 336: ‘precision circa 1 Da’ is not an exact parameter, nor the correct attempt to determine the accuracy or resolution of a MS device. Please change those to an appropriate format.

                Additionally the Methods section lacks of any MS parameters, like ionization setups, mass resolution, ion optic, positive or negative mode, and so on. Those parameters are be mandatory for the analysis of MS data.

line 337: change KDa to kDa

line 344: The mentioned reference [22] gives no additional information in its chapter ‘5.2. Mass Spectrometry and Sequence Determination’ about the process of hydrolysis and should be removed.

line 345: What does the author mean by ‘(mainly  GLU-C,  and  ASP-N)’? What are the other used peptidases, and which concentrations were used?

line 345: What kind of HPLC system and technical setup (column, buffer, gradient, etc.) was used?

line 349: Why ‘usually’? Did the method changed between the different peptides?

data accessibility

The free accessibililty of mass spectrometry data (unprocessed, as well as converted files format like mgf, and others) are important for the reproduction of results. Therefor, the authors should upload those data for the MS1 mass determination on an online repository like ProteomeXChange (https://www.proteomexchange.org) via PRIDE or MassIVE.

line 361: Can the authors please confirm, if really ALL proteins from UniProtKB/Swiss-Prot were used, without any filter.

line 362: What was to original sequence used for the evalue = 1 ×10^(-20) cutoff, Cbo1 or Cbo5; or one of the others?

line 399: Please check manuscript for superscription like here “min−1” wrongly assigned.

Comments on the Quality of English Language

This might be a problems with the used software, but chemical charges are not be superscript (e.g. line 40) and should be controlled manuscript wide. Additionally, why is “chloride” fully written, while the other elements are shown by their abbr.?

as well as in line 399: Please check manuscript for superscription like here “min−1” was wrongly assigned.

Round 2

Reviewer 2 Report

Comments and Suggestions for Authors

The authors changed and corrected a lot of the manuscript, so that it is now clearer to understand and they implemented also several suggestions from the first revision - Thank you for taking my comments into consideration .

At the current state, I recommend minor revisions:

Figure 2 : The figure is now better to understand, only comment here:

line 125/126: the indices “th” and “exp” are not placed low.

Figure 5:

line 237: Figure caption starts with an “_”.

line 239: Maybe change µL min^(-1) to µL/min?

Additionally, please explain the abbr. “HV” again in the image caption.

#####

line 72: It is still not clear in the main text, how the venom was quantified.(former comment to this - line 83/84: “The final content of this component estimated from the initial soluble venom showed that it corresponds to 1.7% of the venom” - How was it determined? Based on the 230nm signal? And if, it lacks the proof, that the isolated fraction were pure without further peptides masses. This could be done by MS1 spectra.)

line 99: Please set the MWth closer to the abbr. explanation of “theoretical molecular weight” in line 97.

Additionally, as in Figure 2 it would be also in the main text good to have “th” and “exp” in the index of MW

line 251: Please mention, that the masses are only correlating with the Cys shift/reduction shift; else it would be an 8 Da difference without explanation.

Comment to Table S1:

Please check the Lad50 abbr. in the table caption above.

It is not clear if the “Supplementary  Table XX. Determinati on  of LD 50  of  C. bonito venom  using the "up and down" “ belongs to S1 or not. And also the calculation seems to be without any number wrongly placed here. Please clarify were this belongs to.

Table caption(?) below is not correct numbered, only with “XX”.

About the MS data:

line 340: The mentioned deconvolution spectra are not shown in the SI, but would be helpful to see directly the peptide masses in addition ti the MS1 spectra. Additionally, please show the ion charges in Figure S1 of the MS1 spectra.

data accessibility again: Even so the MS1 data are now shown in the SI, I still highly recommend to upload the raw data files to an open database as mention in the first Revision round - as stated earlier: “The free accessibililty of mass spectrometry data (unprocessed, as well as converted files format like mgf, and others) are important for the reproduction of results. Therefor, the authors should upload those data for the MS1 mass determination on an online repository like ProteomeXChange (https://www.proteomexchange.org) via PRIDE or MassIVE.”

The registration is free and the upload is easy to handle.

Author Response

ROUN 2

Answer to Reviewer 2

Comments and Suggestions for Authors

“The authors changed and corrected a lot of the manuscript, so that it is now clearer to understand and they implemented also several suggestions from the first revision - Thank you for taking my comments into consideration “.

Dear reviewer, thank you for approving our corrections. Your comments are responded to below, using bold font.

At the current state, I recommend minor revisions:

Figure 2 : The figure is now better to understand, only comment here:

line 125/126: the indices “th” and “exp” are not placed low.

We corrected this issue: “th” and “exp” are now placed low.

Figure 5:

line 237: Figure caption starts with an “_”.

We are sorry, we didn’t find this irregularity in the caption of figure 5

line 239: Maybe change µL min^(-1) to µL/min?

We changed µL min^(-1) to µL/min

Additionally, please explain the abbr. “HV” again in the image caption.

We added the following descrption: “11F, HV, 10FG2 and LR are the arbitrary names for the single chain variable fragment (ScFvs)”

line 72: It is still not clear in the main text, how the venom was quantified.(former comment to this - line 83/84: “The final content of this component estimated from the initial soluble venom showed that it corresponds to 1.7% of the venom” - How was it determined? Based on the 230nm signal? And if, it lacks the proof, that the isolated fraction were pure without further peptides masses. This could be done by MS1 spectra.)

Venom quantification is based on the absorbance at 280 nm (A280) assuming that A280 = 1 corresponds to 1 mg/mL, however the content of each component was estimated as described in this new sentence that we added in the material and methods section, Line 328-329: “The relative concentration of the fractions was estimated from the chromatogram, dividing the area of each peak by the sum of all peaks”.

line 99: Please set the MWth closer to the abbr. explanation of “theoretical molecular weight” in line 97.

Thank you for the suggestion. We accounted for it.

Additionally, as in Figure 2 it would be also in the main text good to have “th” and “exp” in the index of MW

We change “th” and “exp” to the index of MW

line 251: Please mention, that the masses are only correlating with the Cys shift/reduction shift; else it would be an 8 Da difference without explanation.

You are right. Now we add the following in Line 253/255: “The molecular weights found by mass spectroscopy correspond to those expected from the full sequences obtained with a difference of about 8 kD which explain the oxidized state of the cysteines.”

Comment to Table S1:

Please check the Lad50 abbr. in the table caption above.

It is not clear if the “Supplementary  Table XX. Determinati on  of LD 50  of  C. bonito venom  using the "up and down" “ belongs to S1 or not. And also the calculation seems to be without any number wrongly placed here. Please clarify where this belongs to.

Table caption(?) below is not correct numbered, only with “XX”.

Thank you for your advice. We corrected the table S1 in the caption that now states: “Table S1: Determination of LD50 of C. bonito venom using the "up and down" method.”

Below the table we added a brief description of the operations: “From the results reported in the table S1, the LD50 was determined using the formula described by Dixon and Mood:...”

In the equation, we changed “LD50” instead of “m”.

About the MS data:

line 340: The mentioned deconvolution spectra are not shown in the SI, but would be helpful to see directly the peptide masses in addition to the MS1 spectra. Additionally, please show the ion charges in Figure S1 of the MS1 spectra.

data accessibility again: Even so the MS1 data are now shown in the SI, I still highly recommend to upload the raw data files to an open database as mention in the first Revision round - as stated earlier: “The free accessibililty of mass spectrometry data (unprocessed, as well as converted files format like mgf, and others) are important for the reproduction of results. Therefore, the authors should upload those data for the MS1 mass determination on an online repository like ProteomeXChange (https://www.proteomexchange.org) via PRIDE or MassIVE.”

The registration is free and the upload is easy to handle.

Dear reviewer, we understand your point. However, our spectrometer does not have the program for automatically making the deconvolution. We register all the results and then manually determine the average molecular mass of the component.

The raw data will be submitted as suggested.

Additionally, we found and corrected the following mistake:

  • Line 148: changed “...97, 95...” to “...95, 92...” according to the values reported in figure 3